# A Relational-Cultural Approach to Examining Concealment among Latter-Day Saint Sexual Minorities

Samuel J. Skidmore *, Sydney A. Sorrell and Kyrstin Lake

Department of Psychology, Utah State University, Logan, UT 84322, USA; kyrstin.searle@usu.edu (K.L.)
* Correspondence: samuel.skidmore@usu.edu

**Abstract:** Sexual minorities often conceal their sexual identity from others to avoid distal stressors. Such concealment efforts occur more frequently among sexual minorities in religious settings where rejection and discrimination are more likely. Using a sample of 392 Latter-day Saint ("Mormon") sexual minorities, we assess (a) the effect of religious affiliation on concealment efforts, (b) the relationship between social support, authenticity, and religious commitment on concealment, and (c) the moderating effect of authenticity on religious commitment and concealment. Multi-level model analyses revealed that religious affiliation alone accounted for over half (51.7%) of the variation in concealment efforts for Latter-day Saint sexual minorities. Social support directly was related to less concealment, whereas religious commitment was related to more concealment, with authenticity moderating the impact of religious commitment on concealment efforts. The present study provides insight into how religious sexual minorities may approach relationships and inadvertently wound their chances to connect with others.

**Keywords:** LGBTQ; concealment; multi-level modeling; religiousness; relational-cultural theory; Latter-day Saints

## 1. Introduction

Sexual minorities (individuals with some degree of same-sex sexual attraction or behavior) are often faced with external and internal pressures to adhere to heterosexual norms. Such pressures have been linked to a variety of adverse mental health outcomes, including increased stress (Meyer 2003), anxiety, and depression (Escher et al. 2019; Shilo et al. 2016), as well as relationship difficulties including a decrease in perceived social support (Ryan et al. 2009). Many sexual minorities thus opt to conceal their sexual identity from others in an effort to avoid these pressures.

Concealment efforts are most often undertaken in an attempt to protect oneself from potential experiences of prejudice, harassment, and violence from others (Newheiser et al. 2017). Unfortunately, concealment efforts empirically tend to lead to insidious outcomes for sexual minorities; for example, concealment is consistently linked with an increase in ill-being (e.g., depression, anxiety, suicidal thoughts) and a decrease in well-being (e.g., life satisfaction, social belongingness; Livingston et al. 2020). Minority stress theory (Meyer 2003) offers one explanation for this phenomenon; sexual minorities experience stressors that are specific to their marginalized identities in addition to general life stressors, such as work-related concerns or illnesses. Sexual minorities often conceal their sexual identities in an effort to avoid some of these stressors (e.g., prejudice, discrimination). Although sexual minorities who conceal often avoid some degree of external stressors (Livingston et al. 2020), their efforts to conceal are inherently stressful and linked to adverse outcomes such as a sense of disconnection, loneliness, and emotional distress (Kissil and Itzhaky 2015; Shilo et al. 2016).

As part of a natural developmental process, many sexual minorities become attuned to societal pressures to conform to heterosexual norms when they begin to understand

their own identity (Pachankis et al. 2020). Thus, many sexual minorities conceal their identity from others, particularly when they are first coming to terms with their own sexuality (Baiocco et al. 2020). Sexual minorities are more likely to conceal their identity when they believe others will respond negatively to their sexual identity; it is therefore unsurprising that over half of sexual minorities in less-affirming religions conceal their identities, particularly early in their sexual identity development (Shilo et al. 2016; Suen and Chan 2020). Although many sexual minorities ultimately choose to disclose their identity to others (i.e., "come out"), not all follow this trajectory, and those that do sometimes continue to experience the adverse effects of disconnection and social isolation resulting from their concealment (Duncan et al. 2019). Considering the adverse and sometimes lasting effects of concealment on sexual minorities' social and mental health, the current study seeks to elucidate factors influencing concealment efforts among sexual minorities, looking specifically at the effects of social support and religiousness on concealment.

## 2. Understanding Concealment from a Relational-Cultural Theory Perspective

Relational-cultural theory (RCT) is a relatively novel theoretical basis for how sexual minorities may be adversely affected by concealing their identity. RCT is built on the assumption that social disconnection is the issue underlying psychological distress, and that authentic connection with others must be viewed within the context of identities and the culture in which those social connections develop (Jordan 2017). The central relational paradox of RCT posits that human beings crave relationships and belongingness, but often develop "strategies of disconnection" to hide and reject parts of themselves that they see as unlovable or leading to shame. Jordan (2017) specifically noted that authenticity is a necessary component of social connection, and that genuine and beneficial relationships cannot exist without it.

Sexual minorities may be particularly at risk for adverse mental health outcomes when concealing their sexual identity from others. Identity concealment may be understood as a natural attempt to "fit in" with heterosexual individuals and systems, which paradoxically keeps sexual minorities from fully connecting to others. Even when sexual minorities are "out" to their sexual or romantic partners, they may continue to avoid conversations regarding their identity and relationship to others in an effort to avoid social shame or stigmatization. Although often well-intentioned and dependent on context, such identity concealment may be putting their sense of social support and connection at risk (Jordan 2017; Shilo et al. 2016). Indeed, this strategy appears to contribute to an array of difficulties (e.g., depression, anxiety; Kissil and Itzhaky 2015; Livingston et al. 2020). From an RCT perspective, these associations between concealment and adverse mental health outcomes may not be solely due to the difficulties and stress of concealment efforts, but also to the adverse effects of lacking genuine social connection and mutually growth-fostering relationships (Jordan 2017).

Authenticity and social support are vital components to overall well-being and may be threatened by concealment efforts. Authenticity consists of presenting as one's true or core self in day-to-day life (Goldman and Kernis 2002), and is inherently avoided when sexual minorities choose to conceal their identity (Bränström and Pachankis 2021; Newheiser and Barreto 2014). Authentic behaviors are linked to increased social support and connection, which in turn boast a myriad of benefits across aspects of well-being (e.g., increased satisfaction with life, decreased depression; Legate et al. 2012; Solomon et al. 2015). Although the importance of both authenticity and social support on mental health is clear, it is less clear *how* these factors relate to concealment. Concealment efforts are inherently inauthentic and lead to decreased social support, but it is less clear if having a more general sense of authenticity and social support from others can protect against concealment. As such, the present study seeks to better understand how authenticity and social support may affect sexual minorities' concealment efforts.

### 3. Latter-Day Saint Sexual Minorities and Concealment Efforts

In understanding when concealment efforts happen, context must also be taken into account (Rosati et al. 2020). Many sexual minorities conceal their sexual identity from others as a way to avoid potential rejection and discrimination, which both anecdotally and quantitatively occurs more frequently in religious spaces (e.g., Lassiter et al. 2019). Concealment in religious spaces or with religious people often persists due to sexual minorities' concern that, even when they feel a part of the religious community, they will no longer be accepted if they disclose their identity to others. This pattern occurs more frequently in religions that do not affirm same-gender sexual behaviors (e.g., Mainline Protestant, Orthodox Judaism, Latter-day Saints, Islam; Masci and Lipka 2015). Religiously motivated concealment has the same adverse effects as general concealment, with religious sexual minorities reporting greater feelings of isolation, depression, distress, and substance abuse (Escher et al. 2019; Kissil and Itzhaky 2015; Shilo et al. 2016). Religious sexual minorities who ultimately disclose their identities to others also sometimes report increased minority stressors such as discrimination and internalized stigma (Russell and Fish 2016), which may be why many religious sexual minorities continue to conceal their identity despite the adverse outcomes. However, coming out as a religious sexual minority may be a natural part of identity development and boasts a variety of potential benefits, including enhanced social support, increased self-esteem and self-acceptance, and decreased depression (Skidmore et al. 2022b; Vaughan and Waehler 2010). Ultimately, religious sexual minorities may benefit from disclosing their identity to others, including their religious communities, assuming the responses to such disclosure do not include rejection, victimization, or discrimination.

The Church of Jesus Christ of Latter-day Saints (colloquially known as the "Mormon" church) is a conservative Christian denomination that has historically discouraged same-gender sexual behaviors. Although the religion has made shifts away from condemning same-gender *attraction*, members who engage in same-gender sexual behaviors both in and out of marriage are subject to membership restrictions (Church of Jesus Christ of Latter-Day Saints 2016). Latter-day Saint (LDS) sexual minorities often choose to conceal their sexual identity from others, particularly in religious spaces, in an effort to avoid membership consequences in addition to potential discrimination or ostracization (Skidmore et al. 2022b). Such religiously motivated concealment relates to worsened mental health and social isolation (Stammwitz and Wessler 2021), with LDS sexual minorities who are more committed to their religion reporting more concealment of their identity (e.g., Skidmore et al. 2022a). Further, religious commitment relates to an increase in feelings of social support and belongingness among fellow religious family and community members (e.g., Meanley et al. 2016; Skidmore et al. 2022a). The relationship between authenticity and religious commitment is less clear for LDS sexual minorities given the paucity of research in this area. Among the general population, authenticity relates to various aspects of well-being for religious individuals, although such relationships appear to look different across different groups within religions (Ariza-Montes et al. 2017). Although authenticity's relation with well-being is less clear for religious sexual minorities, authenticity seems to largely moderate the relationship between religiousness and a variety of other outcomes, suggesting that authenticity may also moderate how religiousness relates to concealment efforts. Taken together, both religious affiliation generally and religious commitment specifically appear to directly influence concealment efforts for sexual minorities.

### 4. Current Study

In the present study, we build off minority stress theory and relational-cultural theory to elucidate how religiousness and relational-cultural components relate to concealment. Given that many LDS sexual minorities opt to conceal their sexual identity from others, we use a sample of current and former LDS sexual minorities to assess this relationship. We first examine the amount of variance that can be accounted for in concealment by current religious affiliation (RQ1). We then examine whether aspects of religiousness and RCT (religious commitment, authenticity, and social support) affect sexual minorities'

frequency of concealing (RQ2). Based on our review of the literature, we make the following hypotheses:

**H1:** *A substantial proportion of variation in concealment will be accounted for by current religious tradition.*

**H2:** *Social support and authenticity will relate to decreased concealment, whereas religious commitment will relate to increased concealment.*

**H3:** *Authenticity will moderate the relationship between religious commitment and concealment, such that sexual minorities who are more authentic will evidence less concealment even when highly religiously committed.*

## 5. Materials and Method

### 5.1. Procedures and Participants

The research team consisted of six members who represent diverse religious and socio-political identities to manage biases and to recruit a variety of sexual minority and current and former LDS participants. The team represents various identities across religious identity (former LDS, agnostic, Mainline Christian), sexual identity (gay, bisexual, queer, heterosexual, and non-identified), and gender identity (cisgender man, cisgender woman). All members of the research team stand by and uphold the American Psychological Association's position on working with SMs (American Psychological Association 2009).

The research team collected data for the present study from February to March 2022. The Utah State University review board approved all study procedures prior to the start of data collection. Recruitment occurred primarily through community sampling methods. Most participants were recruited from an email list of those who had previously completed the first wave of the research team's longitudinal study and had identified themselves as being interested in completing future surveys (Lefevor et al. 2021). Participants from the first wave of the longitudinal study were recruited via postings in LDS SM forums and social media groups (e.g., Affirmation), advertisements in relevant forums (e.g., the annual North Star conference), therapeutic organizations serving LDS sexual minorities in Utah (e.g., LGBTQ Therapist Guild of Utah), and snowball sampling. In total, 370 interested participants were recruited from the longitudinal study and responded to a series of three emails inviting them to complete a 45-min survey as part of the follow-up study. Additional participants were sought for the present study via similar methods as those employed for the first wave of the longitudinal study. All participants accessed the survey through the research team's website and were offered $10 for participating.

To be included in the study, participants had to be at least 18 years old, identify as a sexual minority (e.g., gay, lesbian, bisexual, queer), be a current or former member of the Church of Jesus Christ of Latter-day Saints, and have completed the entirety of the survey. In total, 392 participants met eligibility criteria and were included in the present study. Many participants identified as cisgender men (54.1%) with a bachelor's degree (43.1%), White (94.6%), gay or lesbian (53.1%), religiously unaffiliated (45.2%), and between the ages of 22–46 ($M$ = 34.41; $SD$ = 12.16). See Table 1 for a full list of participant demographics.

**Table 1.** Demographic Frequencies.

|  | *n* | % |
| --- | --- | --- |
| Race/Ethnicity |  |  |
| Hispanic/Latinx | 2 | 0.5% |
| Native Hawaiian/Pacific Islander | 2 | 0.5% |
| Non-Hispanic White | 371 | 94.6% |
| Multiracial/Other | 17 | 4.4% |
| Education |  |  |
| High school or less | 16 | 4.1% |

**Table 1.** *Cont.*

|  | *n* | % |
|---|---|---|
| Some college | 109 | 27.8% |
| Undergraduate degree | 169 | 43.1% |
| Graduate degree | 98 | 25.0% |
| Gender Identity |  |  |
| Cisgender Man | 212 | 54.1% |
| Cisgender Woman | 104 | 26.5% |
| Transgender Man | 10 | 2.6% |
| Transgender Woman | 10 | 2.6% |
| Gender non-binary/Genderqueer | 46 | 11.7% |
| Other | 10 | 2.6% |
| Sexual Orientation |  |  |
| Gay/Lesbian | 208 | 53.1% |
| Queer | 29 | 7.4% |
| Bisexual | 62 | 15.8% |
| Pansexual | 11 | 2.8% |
| Same-sex/Same-gender attracted | 15 | 3.8% |
| Other | 67 | 17.1% |
| Current Religious Affiliation |  |  |
| Latter-day Saint | 149 | 38.0% |
| Christian-Protestant | 36 | 9.2% |
| Christian-Pentecostal | 5 | 1.3% |
| None | 177 | 45.2% |
| Other | 25 | 6.4% |

*5.2. Measures*

Concealment. Concealment was measured using the Concealment Behavior Scale (Jackson and Mohr 2016). Participants indicated how frequently in the past two weeks they engaged in a series of behaviors related to concealment, such as, "Altered my appearance, mannerisms, or activities in an attempt to 'pass' as straight", and "Allowed others to assume I was straight without correcting them". This scale uses a five-point scale ranging from *not at all* (1) to *all the time* (5). The authors of the study reported that the scale demonstrated excellent reliability and convergent validity (Jackson and Mohr 2016). The scale had a good internal consistency for the present study: α = 0.84.

Religious Commitment. Religious commitment was measured using the Religious Commitment Inventory (Worthington et al. 2003). Participants indicated how true a variety of statements are of them (e.g., spending time trying to grow in their faith, having religious beliefs influence all their dealings in life). The RCI uses a 5-point Likert-type scale ranging from *not at all true of me* (1) to *totally true of me* (5), and has evidenced strong internal consistency, construct validity, and discriminant validity. The scale had excellent internal consistency for the present study: α = 0.94.

Social Support. Social support was measured using the Multidimensional Scale of Perceived Social Support (Zimet et al. 1988). Participants indicated their agreement with a variety of statements using a scale ranging from *strongly disagree* (1) to *strongly agree* (5). Items included statements such as, "I can count on my friends when things go wrong", and "There is a special person in my life who cares about my feelings". The scale demonstrated good reliability and construct validity (Zimet et al. 1988). The scale had excellent internal consistency for the present study: α = 0.91.

Authenticity. Authenticity was measured using Authenticity Scale developed by Wood et al. (2008). Participants indicated how accurately a variety of statements described them using a scale ranging from *does not describe me at all* (1) to *describes me very well* (7). Items included statements such as, "I always stand by what I believe in", and "I think it is better to be yourself, than to be popular". The authors of the scale found support for the measure's reliability as well as construct and concurrent validity (Wood et al. 2008). The scale had good internal consistency for the present study: α = 0.86.

*5.3. Data Analysis*

The distribution of all variables was first examined for normality and showed skewness and kurtosis between −1 and 1. No missing data were identified among the sample. Regression and multilevel analyses were conducted in R version 4.1.2 (R Core Team 2022) using the lme4 package (Bates et al. 2015). To evaluate our hypotheses, we proposed a multilevel modeling regression (also known as repeated measures/mixed effects regression) utilizing current religious affiliation (macrounits; level 2) and individuals (microunits; level 1). In our model, we first utilized a restricted maximum likelihood (REML) null model assessing concealment to determine the intraclass correlation, considering that an ICC of at least 0.05 is typically considered adequate to pursue a multilevel model (Hox et al. 2017). Following this procedure, we utilized a bottom-up approach to multilevel modeling analysis to examine the relationship between Religious Commitment, Social Support, and Authenticity on Concealment, utilizing Religion as the clustering variable. This approach dictates that we add each potential variable to the model individually, first as fixed single effects using maximum likelihood (ML), then as fixed interaction effects using ML, then as random single effects using REML, and finally as random interaction effects using REML. Each model was compared to the last significant model via a likelihood ratio test to determine if the more complex model explains more of the variance in Concealment and should be maintained.

## 6. Results

Descriptive statistics and correlations between variables of interest are presented in Table 2. Overall, participants indicated low levels of Concealment and moderate levels of Religious Commitment, and moderate to high levels of Authenticity and Social Support. Variables related to Relational Cultural Theory (i.e., Authenticity and Social Support) evidenced a moderate degree of intercorrelation ($r = 0.35$), whereas Religious Commitment did not directly correlate with other independent variables.

**Table 2.** Correlations Between Variables of Interest.

|  | *M (SD)* | Range | 1. | 2. | 3. | 4. | 5. |
|---|---|---|---|---|---|---|---|
| 1. Concealment | 1.75 (0.81) | 1–5 |  |  |  |  |  |
| 2. Religious Commitment | 2.38 (1.11) | 1–5 | **0.18** |  |  |  |  |
| 3. Authenticity | 4.89 (0.99) | 1–7 | −**0.46** | 0.04 |  |  |  |
| 4. Social Support | 3.66 (0.83) | 1–5 | −**0.24** | 0.09 | **0.35** |  |  |
| 5. Age | 34.41 (12.16) |  | −**0.16** | **0.10** | **0.24** | −**0.14** |  |

Note: Bolded values indicate relationships significant at $p < $ **0.05**.

To determine if sample characteristics should be included as covariates, demographic variables were grouped into binary categories due to sample size. For example, race/ethnicity (White vs. Person of Color), gender identity (Ciswoman vs. Cisman vs. Transgender/Non-Binary), sexual orientation (Monosexual vs. Plurisexual). Age remained as a continuous variable. Independent samples *t*-tests indicated that race/ethnicity and sexual orientation were not related to any study variable and therefore, not included as controls in our analyses.

Regression analyses indicated that Age was significantly related to Concealment (b = −1.53, *SE* = 0.67, $p < 0.05$), Religious Commitment (b = 1.68, *SE* = 0.59, $p < 0.01$), Social Support (b = −3.32, *SE* = 0.62, $p < 0.01$), and Authenticity (b = 3.31, *SE* = 0.68, $p < 0.01$). Analyses of variance suggested significant differences in Gender (Cisgender vs. Transgender/Gender Non-binary; TGNB) for Concealment, $F = 4.30$, $p < 0.01$, and for Religious Commitment, $F = 6.33$, $p < 0.01$. As such, we dichotomized Gender (Cisgender vs. TGNB) and included this variable as a covariate in our regression analyses alongside Age.

Analysis of the null model demonstrated an ICC of 0.517, indicating that 51.7% of the variance in Concealment scores can be attributed to the group (Religion) level. Concealment scores range on average across different religious affiliations, with Pentecostal

sexual minorities reporting the highest degree of concealment (*M* = 3.53, *SD* = 0.96), followed by Protestant sexual minorities (*M* = 2.46, *SD* = 0.80), then LDS sexual minorities (*M* = 1.83, *SD* = 0.89), sexual minorities from other religions (*M* = 1.61, *SD* = 0.69), and sexual minorities who do not affiliate with any religion (*M* = 1.52, *SD* = 0.57).

See Table 3 for full multilevel modeling regression analyses. Multilevel modeling analyses indicated that, of our covariates, only gender identity significantly related to Concealment (b = 0.31, *SE* = 0.09, *p* < 0.01), suggesting that TGNB sexual minorities are more likely than cisgender sexual minorities to conceal their identities across religions. Of our main study variables, Religious Commitment strongly related to an increase in Concealment (b = 0.77, *SE* = 0.16, *p* < 0.01), whereas Social Support was related to a decrease in Concealment (b = −0.14, *SE* = 0.03, *p* < 0.05). Further, Authenticity significantly moderated the relationship between Religious Commitment and Concealment (b = −0.14, *SE* = 0.03, *p* < 0.01), such that, as Religious Commitment increases, sexual minorities who report high Authenticity also report less Concealment, whereas sexual minorities who report low Authenticity also report more Concealment (See Figure 1).

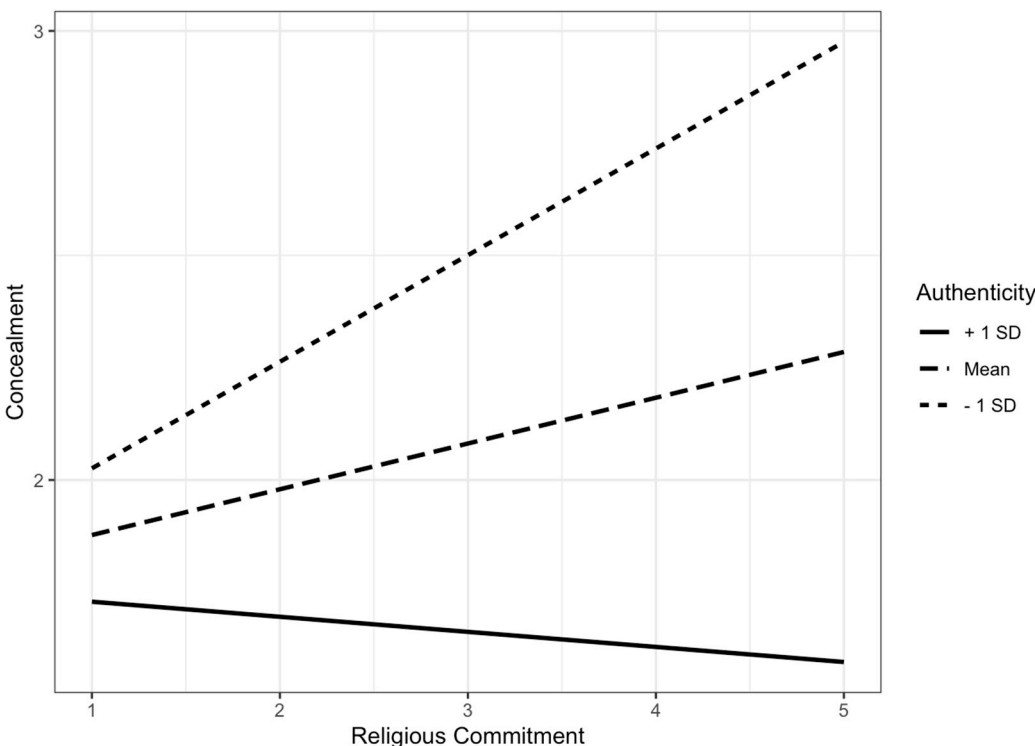

**Figure 1.** The Moderating Effect of Authenticity on Religious Commitment and Concealment.

**Table 3.** Multiple Regression Related to Concealment.

| Study Variables | Null Model | Regression Model | Final Model |
|---|---|---|---|
| **Fixed Effects** | *b (SE)* | *b (SE)* | *b (SE)* |
| Intercept | 2.14 ** (0.31) | 1.87 ** (0.44) | 1.93 ** (0.51) |
| *Control* | | | |
| Age | | 0.01 (0.01) | 0.01 (0.01) |
| TGNB | | 0.35 ** (0.09) | 0.31 ** (0.09) |
| *Main Effect* | | | |
| Religious Commitment | | 0.96 ** (0.17) | 0.77 ** (0.16) |
| Authenticity | | 0.01 (0.08) | −0.01 (0.08) |
| Social Support | | −0.11 * (0.04) | −0.14 * (0.03) |

**Table 3.** *Cont.*

| Study Variables | Null Model | Regression Model | Final Model |
|---|---|---|---|
| *Interaction* | | | |
| Religious Commitment X Authenticity | | −0.16 ** (0.03) | −0.14 ** (0.03) |
| **Random Effects** | | | |
| Religion | 0.44 | | 0.26 |
| **Model Fit** | | | |
| AIC | 899.29 | 801.14 | 832.91 |
| BIC | 911.20 | 813.53 | 849.27 |
| Marginal $R^2$ | 0.517 | 0.298 | 0.516 |

Note. ** $p < 0.01$. * $p < 0.05$.

## 7. Discussion

Findings from the present study suggest that various aspects of religiousness and relational-cultural theory (RCT) are each associated with concealment efforts among a sample of 392 current and former Latter-day Saint (LDS) sexual minorities. Correlations between study variables indicated that participants who reported more religious commitment also reported greater concealment behaviors. In contrast, participants who endorsed a greater degree of authenticity and social support reported decreased concealment efforts. These patterns were further supported by results of subsequent regression analyses. After controlling for the effects of gender and age, multilevel regression analysis revealed that 51.7% of the variance in concealment efforts among participants can be explained by current religious affiliation. Finally, regression analyses revealed that authenticity moderates the relationship between religious commitment and concealment, suggesting that authenticity may act as a protective factor against proximal minority stress for highly religious sexual minorities.

### 7.1. The Influence of Religion on Concealment

We found that a substantial amount of variance in concealment efforts (51.7%) can be attributed to religious affiliation. As expected, not affiliating with any religion was associated with the lowest degree of concealment (Lassiter et al. 2019). In contrast, Pentecostal participants reported the highest degree of concealment, followed by Protestant and current LDS participants. This is likely due to the potential threats of being open about one's sexual identity in a religious environment that does not affirm such identities (Shilo et al. 2016). Sexual minorities who are open about their identities in less-affirming religious contexts often face discrimination, rejection, and even violence (Skidmore et al. 2023; Suen and Chan 2020). Thus, despite the distress typically associated with concealment efforts, the potential threats of openly identifying as a sexual minority in these contexts may seem to outweigh the consequences of concealment for sexual minorities in less-affirming religions. Alternatively, some religious sexual minorities may prioritize a continued sense of belonging to their religious group over authentic expression of their sexual identity, as is common for LDS sexual minorities (Skidmore et al. 2022a). Religion and sexual orientation are each important components of an individuals' identity and navigating potentially conflicting aspects of identity is rarely straightforward. Indeed, almost half of sexual minorities identify as religious, with nearly half of these affiliating with denominations that do not affirm same-gender sexual identities (Williams Institute 2020). While concealment may still be associated with various health risks for sexual minorities in less-affirming religions such as the Church of Jesus Christ of Latter-day Saints (e.g., Legate et al. 2012; Shilo et al. 2016; Skidmore et al. 2022a), it is possible that the threats to belonging to one's religious community are not worth the benefits of being open about one's identity for some sexual minorities.

As expected, religious commitment was consistently associated with greater concealment efforts among participants. Maintaining a sense of belonging to one's religious community may feel more important to sexual minorities who are highly committed to their religion (Skidmore et al. 2022a), potentially even more so than authentic expression of their sexual identity. Furthermore, highly religiously committed individuals tend to be more involved in their religious communities and feel a greater sense of social support and belongingness from religious family and peers (Skidmore et al. 2022a, 2023). As such, highly religiously committed sexual minorities may be more likely to conceal their identities in order to maintain connection with religious peers, who might view their identities as sinful (Lassiter et al. 2019; Rosati et al. 2020). From a minority stress perspective, increased concealment behaviors among highly religious sexual minorities may also result from experiences of internalized stigma, which are more prevalent among sexual minorities in less-affirming religions (Meyer 2003; Masci and Lipka 2015). Highly religiously committed SMs may feel more conflicted about the acceptability of their sexual identities—particularly in the context of a religion that does not affirm sexual minority identities—and may thus be more likely to conceal their identities due to feelings of shame or embarrassment.

Taken together, these findings suggest that a sexual minority's personal commitment to their religion in addition to the specific religion with which they affiliate may influence concealment efforts. Despite the distress and various health risks associated with concealment, concealment may also be a protective factor for highly religious sexual minorities, as it can protect against experiences of discrimination or rejection while also allowing an individual to maintain a sense of belongingness and connection to their religious community (Livingston et al. 2020).

### 7.2. The Impact of Relational Cultural Factors on Concealment

As expected, we found that sexual minorities who reported more support from others also reported decreased concealment behaviors. There are several potential explanations for this phenomenon. RCT posits that social connection contributes to psychological wellness, and that social connections cannot be understood outside the context of identities and the culture in which those social connections develop (Jordan 2017). Therefore, it follows that increased social support for sexual minorities, particularly religious sexual minorities, relates to a decrease in concealment efforts, which are inherently disconnecting. Further, social support relates to increase in outness (Rosati et al. 2020), suggesting that sexual minorities who feel more supported by those around them are more likely to disclose their identity, thereby further increasing connection, support, and authenticity.

Outside of the desire to connect, sexual minorities may engage in less concealment efforts due to the other beneficial effects of social support. Sexual minorities who feel supported by their families and/or peers are better equipped to handle the potential stressors associated with coming out (e.g., discrimination, prejudice, rejection; Li and Samp 2019). These stressors are more prevalent among religious communities (Shilo et al. 2016; Suen and Chan 2020), suggesting that religious sexual minorities may be particularly in need of a sense of social support in order to decrease concealment efforts. Heightened social support may also lend sexual minorities more people to whom they feel close, thus providing additional opportunities to "test" coming out in a way that feels less threatening. Such positive initial coming out experiences lead sexual minorities to be more comfortable and confident in coming out to additional people (Li and Samp 2019). Therefore, sexual minorities who come out first to those with whom they feel supported are more likely to continue to decrease in concealment efforts.

Social support may have further influenced sexual minorities considering its relation to authenticity. Social support and authenticity were moderately correlated to one another, which may explain in part the significant correlation but lack of direct main effect between authenticity and concealment. Jordan (2017) notes that authenticity is a necessary component of social connection, and that genuine and beneficial relationships cannot exist without it. She further notes that authenticity may be put aside in an effort to avoid shame

or stigma from others (e.g., concealment efforts), suggesting that concealment efforts are often intended to protect sexual minorities, but are inherently inauthentic and lead to increased suffering via decreased social connection (Jordan 2017). Sexual minorities who have a strong sense of social support likely feel that they are better able to be authentic in relationships when they already feel supported. Inauthenticity may thus be more likely among people who have yet to shed these strategies of disconnection due to concerns that authentic vulnerability may harm their relationships (Shilo et al. 2016). Taken together, authenticity may be tied with social support with how it influences sexual minorities' concealment efforts; sexual minorities who have a support system are more likely to act authentically (avoid concealment efforts), and sexual minorities who lack a support system are less likely to act authenticity (engage in concealment efforts), thereby maintaining their feelings of disconnection and associated distress (Jordan 2017).

Authenticity further relates to concealment efforts by moderating the way in which religious commitment leads to concealment. Broadly speaking, concealment tends to feel worse for sexual minorities who are more authentic (Bränström and Pachankis 2021; Newheiser and Barreto 2014), likely due to the dissonance associated with wanting to be open regarding oneself but hiding one's sexual identity from others. Thus, for religious sexual minorities who act inauthentically, their commitment to their religion may be leading them to engage in more concealment efforts as a counter-effective strategy to connect with religious others (i.e., central relational paradox; Jordan 2017). Conversely, religious sexual minorities who act more authentically are less likely to engage in concealment efforts, particularly when they are more religiously committed. This opposite effect may be viewed as authentic religious sexual minorities' method for connecting with their religious communities. Indeed, religious sexual minorities who are out to their religious communities experience a myriad of benefits, even when accounting for an increase in distal stressors (e.g., increased self-esteem, self-acceptance, belongingness, and life satisfaction; Li and Samp 2019; Perrin-Wallqvist and Lindblom 2015; Skidmore et al. 2022b; Stuhlsatz et al. 2021). Although all religious sexual minorities may be acting in a way that they believe improves their well-being, it appears that only religious sexual minorities who are most authentic choose not to engage in many concealment efforts, thereby avoiding the array of difficulties that concealment efforts bring (Kissil and Itzhaky 2015; Livingston et al. 2020).

### 7.3. Implications and Limitations

The findings of this research have implications both for research and for informing theory. The current study adds to the growing body of literature demonstrating that, while concealment efforts are undertaken by most sexual minorities, concealment appears to be more common among sexual minorities in religions. Notably, sexual minorities in less-affirming religions and those who are committed to their religion and are less authentic are particularly likely to conceal their identity from others. Future studies may help further elucidate these trends and the associated relationship outcomes associated with such concealment efforts between religions. The current study also demonstrates that authenticity may be a potent influence in how religiousness relates to concealment, such that not all religious sexual minorities choose to conceal their identity, particularly when sexual minorities are more authentic. Given that both concealment and authenticity may change over time, longitudinal studies aimed identifying the ways in which concealment and authenticity change over time for religious sexual minorities may help better define and distinguish these trends.

Findings also have implications for further understanding of theory. RCT is a relatively new theory that specifically notes the importance of context and identity in relationships (Jordan 2017). As such, the present study adds to RCT by demonstrating that sexual minorities—including religious sexual minorities—appear to experience specific threats to connection and authentic relationships, due in part to concealment efforts in order to "fit in' with their religious communities. Further, the present study suggests that authenticity may not directly influence concealment, suggesting that having authentic relationships

with others may not be directly related to whether or not one is concealing their identity broadly speaking, although authentic relationships are likely more difficult to maintain when a sexual minority is concealing their identity from the person with whom they have a relationship.

The present study was limited by several factors relating to sampling and measures. Our sample was drawn from a larger study of current and former LDS sexual minorities. Thus, all participants in this sample were at one point affiliated with this religion, which may have influenced our results. Further, our sample was not evenly distributed between different religious affiliations and is missing many potential affiliations, thus rendering our generalizability limited. The authors are conducting a secondary study using a more extensive distribution of sexual minorities across a variety of religious affiliations in order to improve the generalizability of these results. Finally, our measures were limited in their applicability to RCT. Although authenticity and social support are important components of RCT, we did not have measures regarding other key components of RCT such as growth-fostering relationships and a sense of connection. Additional research is needed in order to analyze these variables among a less biased group of religious sexual minorities to better understand the potential associations and interactions of study variables.

## 8. Conclusions

Using a sample of 392 current and former LDS sexual minorities, we examined the relationships between religiousness and components of relational-cultural theory on SM concealment. Results indicated that religious affiliation alone accounted for over half of the variation in concealment efforts for religious sexual minorities, with sexual minorities in less-affirming religions reported more concealment than sexual minorities in more affirming religions or with no current religion. We further found that social support directly relates to less concealment, whereas religious commitment relates to more concealment. Authenticity moderated the relationship between religious commitment and concealment, such that sexual minorities who were less authentic were more likely to conceal their identity from others when they were more religiously committed, whereas sexual minorities who were more authentic were less likely to conceal their identities from others when they were more religiously committed. These findings provide evidence that a sense of connection to others (via authenticity and social support) decreases concealment, whereas religious affiliation and commitment may increase concealment. The present study has implications both for providing more nuance to the narrative that religious sexual minorities always conceal their identities, as well as providing insight into how religious sexual minorities may approach relationships and inadvertently wound their chances to connect with others.

**Author Contributions:** Conceptualization, S.J.S.; Methodology, S.J.S.; Formal analysis, S.J.S.; Investigation, S.J.S.; Writing—original draft, S.J.S. and S.A.S.; Writing—review and editing, S.J.S., S.A.S. and K.L. All authors have read and agreed to the published version of the manuscript.

**Funding:** This research received no external funding.

**Institutional Review Board Statement:** The study was conducted in accordance with the Declaration of Helsinki, and approved by the Institutional Review Board of Utah State University (approval code 11707, approved on 11/1/2021).

**Informed Consent Statement:** Informed consent was obtained from all subjects involved in the study.

**Data Availability Statement:** Data available upon reasonable request.

**Conflicts of Interest:** The authors report there are no competing interests to declare.

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
