# Peer review of "A Relational-Cultural Approach to Examining Concealment among Latter-Day Saint Sexual Minorities"

_religions, doi:10.3390/rel15020227_

Round 1
Reviewer 1 Report
Comments and Suggestions for Authors
In this manuscript the authors explore aspects of sexual identify concealment among a sample of current or former Mormon sexual minorities.
I think this manuscript is well written and offers an interesting exploration of a component of religious sexual minority experience that tends to be explored using qualitative data analyses rather than using quantitative research methods. The author’s findings that certain groups were more likely to report concealment behaviors than others, those who reported more religious commitment also reported greater concealment behaviors, that authenticity moderates the relationship between religious commitment and concealment, and that sexual minorities who report more support also report less concealment, are important and contribute to the scholarship on religious sexual minorities. I think scholars, practitioners, and parishioners would all be interested in the findings.
The research theory, methods and statistical analysis procedures were clearly described and seemed reasonable, although I did have one question that other readers might also have - do you have to combine trans folks with non-binary/gender queer folks? What happens in the regression analysis if you keep these two groups separate? A few other very minor things - the one sentence paragraph on page 2 seems odd, and I think it is helpful to include the dependent variable in the title of Table 3 rather than the generic “among study variables”.
Author Response
Dear Reviewer,
I sincerely appreciate your thoughtful review of this article. Per your suggestions, we have made the following updates to the manuscript:
- I did have one question that other readers might also have - do you have to combine trans folks with non-binary/gender queer folks? What happens in the regression analysis if you keep these two groups separate?
- Excellent question. We had initially kept these groups separate in as covariates, but in doing so, trans folks vs. other gender identities was nonsignificant, likely due to the sample size being low (n = 20). Given that trans folks and non-binary/genderqueer folks evidenced similar differences compared to other gender identities in preliminary ANOVAs, we opted to combine the groups so that we could still account for all of the variance as opposed to having a nonsignficant covariate in the regression model.
- The one sentence paragraph on page 2 seems odd
- Thank you for noting this. We believe this sentence was left as its own paragraph in error. We have incorporated it into the following paragraph.
- I think it is helpful to include the dependent variable in the title of Table 3 rather than the generic “among study variables”.
- Per your suggestion, we have updated the title of Table 3 to include "Concealment" as the DV.
Again, thank you for your time and thoughtfulness in reviewing this manuscript! We firmly believe the associated changes have strengthened the paper and will make for a more compelling read.
Reviewer 2 Report
Comments and Suggestions for Authors
This is a straightforward, solid empirical contribution to the literature on the subject.
A stylistic note: Lines 70-72 is a one-sentence paragraph that does not have a good reason to be a one-sentence paragraph; in that case it might be better to merge it with the preceding or subsequent paragraph.
"The religion no longer explicitly condemns same-gender attraction, behaviors, or relationships, although members who engage in same-gender sexual behaviors both in and out of marriage are subject to membership restrictions (Church of 124 Jesus Christ of Latter-day Saints, 2016)."
It seems a little confusing to argue that they do not condemn same-sex relationships or behaviors, when mentioning in the same line that there are membership restrictions for those who engage in same-sex sexual behavior.
I'm a little uncomfortable about the implication in the background section that (e.g. lines 74-77) that if, for example, somebody decides not to be out to their great-grandma, that they are ipso facto “inauthentic,” I get the point that the authors are making, but how and in what context somebody is out is highly personal and can be complex, and it might be better to avoid using a pejorative label for everybody who isn’t 100% out to everybody in their life.
On top of that, connecting the two by definition makes it more confusing later on when the authors operationalize the two as distinct concepts and then empirically demonstrate that they are interrelated. It seems like it would be more straightforward to stick to defining the two distinctly, and then, as is already well done in the paper, theorize about why the two distinct concepts could be expected to be empirically related.
Author Response
Dear Reviewer,
We sincerely appreciate your thoughtful review of this article. Per your suggestions, we have made the following updates to the manuscript:
- A stylistic note: Lines 70-72 is a one-sentence paragraph that does not have a good reason to be a one-sentence paragraph; in that case it might be better to merge it with the preceding or subsequent paragraph.
- Thank you for noting this. We believe this sentence was left as its own paragraph in error. We have incorporated it into the following paragraph.
- "The religion no longer explicitly condemns same-gender attraction, behaviors, or relationships, although members who engage in same-gender sexual behaviors both in and out of marriage are subject to membership restrictions (Church of 124 Jesus Christ of Latter-day Saints, 2016)." It seems a little confusing to argue that they do not condemn same-sex relationships or behaviors, when mentioning in the same line that there are membership restrictions for those who engage in same-sex sexual behavior.
- That is a very valid point. We have received feedback in that past that the CJCLDS has made statements that same-gender attraction and behaviors are no longer "condemned" in terms of not necessitating excommunication from the religion, although there remain membership restrictions. We have updated this sentence in the manuscript so that this is clearer.
- "Although the religion has made shifts away from condemning same-gender attraction, members who engage in same-gender sexual behaviors both in and out of marriage are subject to membership restrictions (Church of Jesus Christ of Latter-day Saints, 2016)."
- I'm a little uncomfortable about the implication in the background section that (e.g. lines 74-77) that if, for example, somebody decides not to be out to their great-grandma, that they are ipso facto “inauthentic,” I get the point that the authors are making, but how and in what context somebody is out is highly personal and can be complex, and it might be better to avoid using a pejorative label for everybody who isn’t 100% out to everybody in their life. On top of that, connecting the two by definition makes it more confusing later on when the authors operationalize the two as distinct concepts and then empirically demonstrate that they are interrelated. It seems like it would be more straightforward to stick to defining the two distinctly, and then, as is already well done in the paper, theorize about why the two distinct concepts could be expected to be empirically related.
- We really appreciate this point, as we did not have the intention to shame individuals who choose to conceal their sexual identity from others, or suggest that it is inherently a harmful thing. We have updated the language here to rectify this error (see below):
- "Even when sexual minorities are “out” to their sexual or romantic partners, they may continue to avoid conversations regarding their identity and relationship to others in an effort to avoid social shame or stigmatization. Although often well-intentioned and dependent on context, such identity concealment may be putting their sense of social support and connection at risk (Jordan, 2017; Shilo et al., 2016)."
Again, thank you for your time and thoughtfulness in reviewing this manuscript! We firmly believe the associated changes have strengthened the paper and will make for a more compelling read.